# Traffic-Data Recovery Using Geometric-Algebra-Based Generative Adversarial Network

**DOI:** 10.3390/s22072744

**Published:** 2022-04-02

**Authors:** Di Zang, Yongjie Ding, Xiaoke Qu, Chenglin Miao, Xihao Chen, Junqi Zhang, Keshuang Tang

**Affiliations:** 1Department of Computer Science and Technology, Tongji University, Shanghai 200092, China; dingyongjie2000@163.com (Y.D.); quxiaoke@tongji.edu.cn (X.Q.); 1851804@tongji.edu.cn (C.M.); 2030803@tongji.edu.cn (X.C.); zhangjunqi@tongji.edu.cn (J.Z.); 2Department of Transportation Information and Control Engineering, Tongji University, Shanghai 200092, China

**Keywords:** traffic data recovery, geometric algebra, deep learning, intelligent transportation system

## Abstract

Traffic-data recovery plays an important role in traffic prediction, congestion judgment, road network planning and other fields. Complete and accurate traffic data help to find the laws contained in the data more efficiently and effectively. However, existing methods still have problems to cope with the case when large amounts of traffic data are missed. As a generalization of vector algebra, geometric algebra has more powerful representation and processing capability for high-dimensional data. In this article, we are thus inspired to propose the geometric-algebra-based generative adversarial network to repair the missing traffic data by learning the correlation of multidimensional traffic parameters. The generator of the proposed model consists of a geometric algebra convolution module, an attention module and a deconvolution module. Global and local data mean squared errors are simultaneously applied to form the loss function of the generator. The discriminator is composed of a multichannel convolutional neural network which can continuously optimize the adversarial training process. Real traffic data from two elevated highways are used for experimental verification. Experimental results demonstrate that our method can effectively repair missing traffic data in a robust way and has better performance when compared with the state-of-the-art methods.

## 1. Introduction

Traffic data are of great significance to intelligent transportation systems (ITS), which provide useful information for traffic flow prediction, congestion judgment and urban transportation network planning. Accurate traffic data can make the analysis results more reliable. In a large-scale traffic-flow-monitoring system, sensors deployed in different locations can collect a large amount of useful time series data. However, due to the influence of the hardware device itself, the sensors often fail to work, resulting in incomplete data collection [1]. At the same time, accidents that occur during the storage of a large amount of traffic data will also cause the lack of these traffic data. In order to repair the missing traffic data, researchers have tried a variety of methods including regression-model-based methods, probability-model-based methods and deep-learning-based methods.

## 2. Related Work

Regression-model-based methods evaluate the mathematical expectations of missing data through known data points. Local binary pattern (LBP)-based support vector machines (SVMs) [2] have shown better recovery results when a small amount of traffic data are missing. Least squares support vector machines (LS-SVMs) introduced by Zhang and Liu [3], and the K-value proximity algorithm based on spatial and temporal correlation [4], also illustrate better imputation performance when missing types and data are mixed. Online support vector machines (OL-SVR) proposed by Manoel [5] have more timely responses in repetitive traffic data. However, most regression models cannot recover data with high signal-to-noise ratio (SNR) or long sequences of missing data, which often occurs in ITS systems [6].

Probability models include the principal component analysis (PCA) [7] method, based on historical data mining, and the fully Bayesian generative model [8], based on tensor decomposition for estimating missing data. Bayesian principal component analysis (BPCA) [9] combines these two algorithms mentioned above to achieve a balance between the periodicity of the flow, local predictability, and statistical properties of traffic. The Bayesian Gaussian CANDECOMP/PARAFAC tensor decomposition (BGCP) [10] algorithm extends the tensor decomposition to higher dimension and applies it to the spatio–temporal traffic data interpolation task, solving the problem of missing data attribution in a spatio–temporal multidimensional environment. The variational Bayesian (VB) [11] algorithm exploits the spatio–temporal properties of network traffic to improve the quality of lost data recovery, fully capturing the multidimensional and spatio–temporal characteristics of traffic data.

Deep learning has demonstrated its great potential in many fields, including transportation [12]. Deep-learning-based data recovery models rely on high scale traffic data and incorporate the influence of nonlinear factors in a better way. Convolutional neural networks (CNNs) [13] are commonly used for image data recovery and improving image resolution, and super-resolution convolutional neural networks (SRCNNs) proposed by Dong et al. [14] can learn the recovery process from low-resolution images to high-resolution images in an end-to-end manner.

Generative adversarial networks (GAN) [15] are generative models which can create new data instances that resemble the training samples, they have been widely applied in many domains such as image restoration [16], video prediction [17] and security [18]. GANs are also used for traffic information recovery by using historical traffic data to improve recovery accuracy. He and Luo et al. [19] propose the research of GAN in traffic-data recovery. Arora, S. [20] conducts a study on the generalization ability of GAN in different situations. The encoded multiagent generative adversarial network (E-MGAN) proposed by Zhao [21] proves to be very effective in overcoming GAN pattern collapse. Deep Convolutional Generative Adversarial Network (DCGAN) [22] and Generalized Adversarial Interpolation Network (GAIN) [23] can solve the model instability problem to a certain extent. M. Arif [24] establishes a deep learning model with nonparametric regression to improve the prediction of lost data under nonlinear spatio–temporal effects. D. Tran et al. [25] finds that 3D convolution is more suitable for spatio–temporal feature learning than 2D convolution, easier to train and use. K. Xie [26] proposes a sequential tensor completing method to reduce the computing cost of high-dimensional neural network algorithms. All the above studies have promoted the application of 3D convolutional generative adversarial networks [27,28] that can effectively recover traffic data in large-scale traffic networks.

In summary, existing research has made some progress in the field of traffic-data restoration. But the accuracy of repairing large-scale missing data still needs to be improved. Traffic data are composed of multiple parameters, such as flow, speed and occupancy. These parameters are interrelated and contain complex high-dimensional traffic laws. Geometric algebra has strong expressive ability for multidimensional signals, and can better realize the learning of high-dimensional correlation. In this paper, considering the advantages of deep learning, we propose a geometric-algebra-based generative adversarial network (GAGAN) to recover missing traffic data by learning the correlation of multidimensional traffic parameters. The performance of traffic-data repair can be improved by coupling geometric algebra and generative adversarial network into a single model. We first preprocess the original traffic data, which include speed, flow and occupancy, to generate scalar-valued spatio–temporal matrices. By embedding the traffic data in the framework of geometric algebra, multivector-valued spatio–temporal matrices, which contain elements of high-dimensional entities, are created and used as the inputs of the proposed GAGAN model. The generator of GAGAN consists of a geometric algebra convolutional module, an attention module and a deconvolutional module. The discriminator of GAGAN is composed of a multichannel convolutional neural network.

The main contributions of this paper are summarized as follows:We present a geometric algebra based generative adversarial network (GAGAN) to handle the problem of traffic data recovery. To represent and process multidimensional signals more efficiently, original traffic data are embedded in the framework of geometric algebra to form multivector-valued spatial-temporal matrices.The generator of the proposed GAGAN contains a geometric algebra convolutional module, an attention module and a deconvolutional module. The geometric algebra convolutional module is capable of learning the correlations of multidimensional inputs more efficiently. The loss function of the generator considers both the global and local traffic data mean squared errors.We conduct various experiments based on traffic data from two urban expressways of Shanghai, China. Experimental results prove that our method can effectively repair missing traffic data in a robust way. Compared with the state-of-the-art work, our approach shows the best performance.

## 3. Geometric Algebra of Euclidean 3D Space

Geomtric algebra [29,30] is a generalization of vector algebra and it has been succesfully applied in the domain of physics and engineering [31]. Compared with the classical vector algebra, modeling capability based on geometric algebra is tremendously extended. As a coordinate-free system, it captures the geometric characteristics of the problem in a better way and enables a more powerful representation and processing framework for multidimensional signals. Since the traffic data recovery problem is handled in the 3D Eulidean space ($R^{3}$), in this section, we breifly introduce the geometric algebra of Euclidean 3D space ($R_{3}$).

As shown in Equation (Equation 1), there are 8 basis elements of the gemetric algebra of 3D Euclidean space ($R_{3}$).
(1)$$R_{3}=span\left\{1,e_{1},e_{2},e_{3},e_{12},e_{23},e_{31},e_{123}\right\}$$
where 1 indicates the scalar basis, $e_{1}$, $e_{2}$ and $e_{3}$ refer to orthonormal basis vectors; $e_{12}$, $e_{23}$ and $e_{31}$ indicate unit bivectors; $e_{123}$ means the unit trivector.

For a unit cube, $e_{1}$, $e_{2}$ and $e_{3}$ represent three axes, $e_{12}$, $e_{23}$ and $e_{31}$ correspond to three surfaces and $e_{123}$ indicates the cube. By combining these basis elements, a multivector can be formed to represent multidimensional entities in an efficient way, e.g., $M=3+5e_{1}+7e_{2}+9e_{3}+11e_{12}+13e_{23}+15e_{31}+17e_{123}$. Geometric product is the basic product of geometric algebra, it is noncommutative and can be decomposed as the combination of inner product and outer product, Table 1 shows the results of geometric products of basis elements.

Given two multivectors $M_{1}=3e_{1}+5e_{23}$ and $M_{2}=3e_{2}+7e_{12}$, there geometric product is given by
(2)$$M_{1}⊗M_{2}=M_{1}\cdot M_{2}+M_{1}\wedge M_{2}=21e_{2}-15e_{3}+9e_{12}+35e_{31}$$
where ⊗, · and ∧ represent geometric product, inner product and outer product, respectively.

## 4. Proposed Methodology

### 4.1. Overview

This paper aims to realize the repair of damaged traffic data. Figure 1 shows our system architecture. Raw traffic data which include speed, flow and occupancy are collected by the detectors deployed on the elevated highway at specific time intervals, and there is a position interval between these detectors. First, raw traffic data are preprocessed, and then converted into spatio–temporal matrices, each of which integrates certain traffic information of a day in both spatial and time domains. The matrix containing speed information is used to generate damaged speed matrix using point-by-point multiplication with a randomly generated mask of the same size. Next, the damaged speed matrix, the complete flow matrix and the complete occupancy matrix become a sample whose label is the complete speed matrix. Samples of all days constitute a data set. We randomly divide the samples into training samples to train proposed GAGAN model and test samples to repair and test the performance of our model. Recovered speed matrix is obtained by multiplying predicted speed matrix generated by GAGAN with the mask-inverted matrix.

### 4.2. Damaged Data Set Generation

Traffic data are collected by detectors deployed on the road. Different roads have different value ranges for the same traffic parameter. Therefore, it is necessary to normalize the traffic data including flow, speed and occupancy. For example, the regularization of speed can be described as:(3)$$s_{norm}=\frac{s-s_{min}}{s_{max}-s_{min}}$$

$s_{norm}$ represents normalized data while s is original speed data. $s_{max}$ means the maximum value of the original speed data, and $s_{min}$ is the minimum value. Flow and occupancy are also processed in the same way.

Because the detectors are deployed at different locations on the road and collect traffic data at regular intervals, the traffic data itself has time and space properties. In order to make full use of the correlation between time and space, we construct the traffic spatial–temporal matrix. A row of the matrix indicates the location of a detector, and different columns represent different times of a day. The matrix elements refer to values of traffic speed. Mathematically, traffic speed spatial–temporal matrix can be represented as:(4)$$S=\left[\begin{array}{cccc} S_{11} & S_{12} & ... & S_{1n}\\ S_{21} & S_{22} & ... & S_{2n}\\ ... & S_{ij} & ... & ...\\ S_{m1} & S_{m2} & ... & S_{mn} \end{array}\right]$$

The matrix *S* represents the traffic speed information for each day. Where *m* and *n* are the number of loop detectors and the number of time intervals respectively, $S_{ij}$ is normalized speed of the $i_{th}$ loop detector at the $j_{th}$ time period. Similarly, we can get the flow spatial–temporal matrix and occupancy spatial–temporal matrix, which are represented as F and O, respectively.

Next, we simulate the damage to the traffic speed data. Traffic data corruption usually occurs in various locations. Moreover, the shape of the damaged part is also different. Therefore, we use two different shapes of masks to randomly destroy the data. One is the strip damage, in this case damaged data is continuous with time, which in the space–time matrix is displayed as a rectangle. The other is the discrete damage, that means the damaged data is discontinuous, which in the space–time matrix is displayed as dots. Mathematically, the mask can be defined as:(5)$$Mask=\left[\begin{array}{cccc} k_{11} & k_{12} & ... & k_{1n}\\ k_{21} & k_{22} & ... & k_{2n}\\ ... & k_{ij} & ... & ...\\ k_{m1} & k_{m2} & ... & k_{mn} \end{array}\right]$$
where the value of $k_{ij}$ is 0 or 1. If it is 0, the data of this point is damaged. If it is 1, the data of this point is retained.

Finally, we multiply the speed spatial–temporal matrix and the mask point-by-point to obtain a corrupted data set.

### 4.3. The GAGAN Model for Traffic Speed Recovery

The GAN model has been proved to perform very well in the application of image generation, and geometric algebra has the advantage of representing and processing multidimensional signals in an efficient way; therefore, we are inspired to propose the GAGAN model for traffic speed recovery by coupling GAN and geometric algebra into a single framework.

As shown in Figure 2, three scalar-valued matrices, i.e., damaged speed, complete flow and occupancy, are employed as the input of GAGAN model. By embedding these scalar-valued matrices in the gemoetric algebra, a multivector-valued matrix which represents multidimensional signals can be obtained and considered as the input of the generator. The generator of GAGAN is a geometric algebra convolutional neural network (GACNN) with multivector-valued neurons; it aims to learn the correlation of multidimensional traffic data and generate a recovered speed matrix. The discriminator of GAGAN contains a scalar-valued multichannel CNN, which is applied to determine whether the result generated by the generator is true or false, and to continuously feed back information to the generator, thereby improving the model’s repair accuracy.

Even though the presented GAGAN model in this paper is used to recover missing traffic speed data, it also can be generalized to recover different types of data based on multidimensional inputs.

#### 4.3.1. The Generator of GAGAN

The structure of the generator is a GACNN, as illustrated in the Figure 3. It consists of two parts: encoding and decoding. The encoding part includes 3 geometric algebra convolutional layers, 3 pooling layers and 1 convolutional block attention module (CBAM). The function of the encoding part is to produce advanced feature maps which can efficiently describe the correlation characteristics of the input. The decoding part of the generator consists of 3 deconvolutional layers, aiming to decode the comprehensive spatio–temporal features extracted from the traffic parameters, and output the repaired speed matrix with the same size as the input speed matrix. Compared with scalar-valued CNN, GACNN has better capability to learn the potential dependencies between mutidimensional inputs.

The orignial inputs of the GAGAN model are damaged speed, complete flow and occupancy matrices of a day, they are first embedded in the geometric algebra with bivector basis to yield a multivector valued matrix as the input of the generator, which can be represented as Equation (Equation 6).
(6)$$Z=\left[\begin{array}{cccc} Z_{11} & Z_{12} & ... & Z_{1n}\\ Z_{21} & Z_{22} & ... & Z_{2n}\\ ... & Z_{ij} & ... & ...\\ Z_{m1} & Z_{m2} & ... & Z_{mn} \end{array}\right]$$
where the matrix ***Z*** indicates the multivector valued spatio–temporal matrix which encodes the traffic information for a day, *m* and *n* are the number of loop detectors and the number of time intervals respectively, $Z_{ij}$ is the multivector valued traffic parameter of the $i_{th}$ loop detector at the $j_{th}$ time period, $Z_{ij}$ can be further expressed in the following form:(7)$$Z_{ij}=F_{ij}e_{12}+S_{ij}e_{23}+O_{ij}e_{31}$$
where $F_{ij}$, $S_{ij}$ and $O_{ij}$ refer to the flow, speed and occupancy, respectively.

The geometric algebra convolutional layers of the generator are able to extact coorelated spatio–temporal features by convolving the input with learnable kernels. Different with the conventional scalar-valued convolution, in this case, both the input and kernel are multivector-valued. For the $L_{th}$ geometric algebra convolutional layer, the input of the multivector-valued neuron is the output of the previous layer, which can be denoted as
(8)$$X_{ij}^{L-1}=X_{ij}^{L-1,r}+X_{ij}^{L-1,1}e_{12}+X_{ij}^{L-1,2}e_{23}+X_{ij}^{L-1,3}e_{31}$$
where $X_{ij}^{L-1}$ means the output of the previous layer, r indicates the scalar part of the multivector $X_{ij}^{L-1}$, 1, 2, 3 represent the corresponding bivector parts.

For the first layer, since there are only 3 traffic parameters, the scalar part of $X_{ij}^{L-1}$ is zero, i.e., $X_{ij}^{L-1}=Z_{ij}$. However, according to the results of geometric product, $X_{ij}^{L-1}$ at other layers will contain scalar parts. To perform the geometric algebra convolution, the weights $W_{ij}^{L}$ of the kernel in the $L_{th}$ layer also take multivector values as shown in Equation (Equation 9)
(9)$$W_{ij}^{L}=W_{ij}^{L,r}+W_{ij}^{L,1}e_{12}+W_{ij}^{L,2}e_{23}+W_{ij}^{L,3}e_{31}$$

Hence, the convolved output of a neuron in the $L_{th}$ geometric algebra convolution layer reads
(10)$$\begin{array}{cc} X_{ij}^{L} & =f\left(\sum_{i=1}^{p}\sum_{j=1}^{q}X_{ij}^{L-1}⊗W_{ij}^{L}+B_{ij}^{L}\right)\\ \; & =f\left(\sum_{i=1}^{p}\sum_{j=1}^{q}\left(X_{ij}^{L-1}\cdot W_{ij}^{L}+X_{ij}^{L-1}\wedge W_{ij}^{L}\right)+B_{ij}^{L}\right) \end{array}$$
where f is the ReLU activation function, the kernel has a size of p×q, ⊗, ·, ∧ respectively represent geometric product, inner product and outer product, $B_{ij}^{L}$ means the bias parameter of this layer. According to the relationship shown in Table 1, the geometric product of two multivectors $X_{ij}^{L-1}$ and $W_{ij}^{L}$ is defined as:(11)$$\begin{array}{cc} X_{ij}^{L-1}⊗W_{ij}^{L}= & X_{ij}^{L-1}\cdot W_{ij}^{L}+X_{ij}^{L-1}\wedge W_{ij}^{L}\\ = & D_{r}+D_{1}e_{12}+D_{2}e_{23}+D_{3}e_{31} \end{array}$$
where $D_{r}$, $D_{1}$, $D_{2}$ and $D_{3}$ are scalar coefficients which can be further expressed as:(12)$$\begin{array}{cc} D_{r}= & X_{ij}^{L-1,r}W_{ij}^{L,r}-X_{ij}^{L-1,1}W_{ij}^{L,1}\\ \; & -X_{ij}^{L-1,2}W_{ij}^{L,2}-X_{ij}^{L-1,3}W_{ij}^{L,3} \end{array}$$
(13)$$\begin{array}{cc} D_{1}= & X_{ij}^{L-1,r}W_{ij}^{L,1}+X_{ij}^{L-1,1}W_{ij}^{L,r}\\ \; & +X_{ij}^{L-1,2}W_{ij}^{L,3}-X_{ij}^{L-1,3}W_{ij}^{L,2} \end{array}$$
(14)$$\begin{array}{cc} D_{2}= & X_{ij}^{L-1,r}W_{ij}^{L,2}-X_{ij}^{L-1,1}W_{ij}^{L,3}\\ \; & +X_{ij}^{L-1,2}W_{ij}^{L,r}+X_{ij}^{L-1,3}W_{ij}^{L,1} \end{array}$$
(15)$$\begin{array}{cc} D_{3}= & X_{ij}^{L-1,r}W_{ij}^{L,3}+X_{ij}^{L-1,1}W_{ij}^{L,2}\\ \; & -X_{ij}^{L-1,2}W_{ij}^{L,1}+X_{ij}^{L-1,3}W_{ij}^{L,r} \end{array}$$

The geometric algebra convolution layer is mainly based on the geometric product operation to realize the information transfer between multivector neurons. The neurons are connected locally and the weights are shared. From Equations (Equation 10)–(Equation 15), it is demonstrated that the traditional scalar-valued convolution, indicated by the inner product, is just a part of the geometric algebra convolution. In addition, the geometric algebra convolution includes the computation of outer products, it provides the potential to learn the correlations of multidimensional inputs. Compared with 3D convolution, which ignores the relationship between channels and causes information loss, the geometric algebra convolution is capable of learning coorelation features of multidimensional signals in a more efficient way.

Geometric algebra is the basic mathematical framework to model our problem, however, in the real implementation, we follow the way illustrated in Figure 4 to perform the computation of a geometric algebra convolutional layer L. For the geometric product, we map multivector-valued neurons to multiple scalar neurons according to the number of dimensions. In this case, one multivector-valued neuron corresponds to four scalar-valued neurons, their outputs can be obtained according to equations from (Equation 11) to (Equation 15) by adding and subtracting the results of 4 ordinary convolutions. The geometric algebra convolution is similar to learning the compound characteristics by aggregating several separate standard convolution results. The four convolved results are then combined by basis to form the multivector-valued input for the next layer.

It is worth noting that for the area to be repaired, the traffic area far away from it does not provide much information and may even interfere with the repair result. We introduce the convolutional block attention module (CBAM) to extract useful information and filter useless information, thereby improving feature extraction capabilities. Mathematically, the process of CBAM can be defined as:
(16)$$\begin{array}{cc} F_{c}=\sigma ( & FC\left(\text{AvgPool}\left(F_{\mathrm{ga}}\right)\right)+\\ \; & \text{FC}\left(\text{MaxPool}\left(F_{ga}\right)\right))⨀F_{ga} \end{array}$$
(17)$$\begin{array}{c} F_{S}\left(F_{c}\right)=\sigma (f^{7\times 7}(\text{AvgPool}\left(F_{c}\right)⨁\\ \text{MaxPool}\left(F_{c}\right)))⨀F_{c} \end{array}$$

$F_{ga}$, $F_{c}$ and $F_{S}$ represent feature maps obtained from geometric algebra convolution layer, channel attention module and spatial attention module, respectively, AvgPool and MaxPool represent average pooling and maximum pooling, respectively, FC and $f^{7\times 7}$ refer to the fully connected layer and convolutional layer using the convolution kernel with a size of 7 × 7, σ denotes the sigmoid function, ⨀ represents point-by-point multiplication between matrices, ⨁ denotes concatenating channels.

The CBAM layer is composed of two parts, first the channel attention module, and then the spatial attention module.The channel attention module first adopts global average pooling and global maximum pooling. Then the feature maps are delivered to the fully connected layers to model the correlation between the channels. The weight of the feature channel is defined in Formula (Equation 16) as the part between symbols of = and ⨀, they are multiplied channel-by-channel to complete the recalibration of the original feature in the channel dimension. The spatial attention module takes the output of the channel attention module as the input. The global average pooling and global maximum pooling are also used. The difference is that the pooling operation compresses the multichannel feature map into a single channel, so that the subsequent convolution only focuses on the spatial dimension. Finally, it is the same recalibration operation that the newly obtained weight in the spatial dimension is multiplied by the feature map to yield the result adjusted by double attention models.

After completing the extraction of the high-dimensional features of the traffic parameters, the extracted high-dimensional features need to be decoded. As mentioned above, we perform feature encoding based on three bivectors and one scalar. For decoding, deconvolution is performed for the four dimensions of three bivectors and one scalar. The feature maps obtained by deconvolution decode the traffic speed information layer by layer. Finally, we fuse the feature maps generated from these four dimensions, stitch them together according to the channels, and then pass them to the last deconvolution layer to produce the recovered traffic speed matrix.

#### 4.3.2. The Discriminator Structure

The discriminator of GAGAN can be regarded as a binary classifier, it aims to distinguish as accurately as possible whether the input is the ground truth or the recovered value yielded by the generator. The discriminator fights against the generator, which further encourages the generator to produce more realistic recovered values. It has been proved that the performance of GAN will be improved if it is conditioned. The proposed GAGAN model is a conditional GAN, as illustrated in Figure 5. The discriminator is composed of multiple CNNs, each CNN consists of 2 convolutional layers, 2 pooling layers and 2 fully connected layers. Multidimensional data including flow, occupancy and damaged speed are taken as conditions and fed to three CNNs for learning patterns and distributions, features of the predicated speed matrix is also learned by another CNN. Concatenating results from these four CNNs produces $P_{1}$, a value which indicates the probability of the output of generator coming from training samples. In addition to this, the ground truth of speed matrix is also delivered to the fifth CNN, and $P_{2}$, the probability of real values coming from training samples can be obtained. In this paper, the multiple conditions applied to our model enable the discriminator to make a more reliable decision of the probability that the predicted value is the true value under the constraints of the current FSO matrices.

#### 4.3.3. Model Optimization

Model training is a process of continuously adjusting the weight parameters. The model is composed of the generator network (G) and the discriminator network (D), they compete with each other and are trained alternately.

The goal of the discriminator is to distinguish as accurately as possible whether the input is the data generated by the generator or the real data, by minimizing the probability $P_{1}$ and maximizing the probability $P_{2}$, as shown in Figure 5. Thus, the loss function of D is the crocess entropy, which can be defined as:(18)$$L_{D}=-\log\left(1-D\left(G\left(\hat{C}\right)\right)\right)-\log\left(D\left(x|\hat{C}\right)\right)$$
where $\hat{C}$ means the FSO condition matrices, $G(\hat{C})$ indicates the output generated by the generator, $D(G\left(\hat{C}\right))$ denotes the probability $P_{1}$, x refers to the training data coming from real distribution and $D(x|\hat{C})$ denotes the probability $P_{2}$.

The goal of the optimization is to make the repaired value, that is the output of the generator, as close to the real value as possible. Based on the traditional GAN [32], we optimize the loss function of the generator as
(19)$$L_{G}=\alpha L_{g}+\beta L_{\text{totalMSE}}+\gamma L_{\text{localMSE}},$$
with
(20)$$L_{g}=-\log\left(D\left(G\left(\hat{C}\right)\right)\right)$$
(21)$$L_{\text{totalMSE}}=\frac{1}{N}\sum_{t=1}^{N}\left(\frac{1}{mn}\left(\sum_{i=1}^{m}\sum_{j=1}^{n}{\left(S_{ijt}-{\hat{S}}_{ijt}\right)}^{2}\right)\right)$$
(22)$$\begin{array}{cc} L_{localMSe}= & \frac{1}{N}\sum_{t=1}^{N}(\frac{1}{C_{t}}(\sum_{i=1}^{m}\sum_{j=1}^{n}\\ \; & {\left(\left(S_{ijt}-{\hat{S}}_{ijt}\right)*\left(1-{\text{Mask}}_{ijt}\right)\right)}^{2})) \end{array}$$
where α, β, and γ, whose sum equals 1, are weights associated with 3 parts of the loss function of GAGAN. $L_{g}$ is used to measure the authenticity of the generated results and make the generated value from G more approximate to the real value. $L_{\mathrm{totalMSE}}$ represents the global mean squared error (MSE), it measures the overall loss between the speed matrix generated by the generator and the real matrix. N refers to the number of test samples, m and n are the number of rows and columns of a speed matrix, respectively. $S_{ijt}$ denotes the true speed value of the the $i_{th}$ loop detector at the $j_{th}$ time period of the $t_{th}$ test sample, ${\hat{S}}_{ijt}$ means the corresponding recovered value. $L_{\text{localMSE}}$ is used to measure the loss between the recovered value and true value in the damaged area, so as to learn the characteristics of the damaged area in a targeted manner. $C_{t}$ indicates the number of damaged points in the $t_{th}$ speed matrix. Similar to $S_{ijt}$, ${\text{Mask}}_{ijt}$ means the mask value of the the $i_{th}$ loop detector at the $j_{th}$ time period of the $t_{th}$ test sample. The multiplier ‘$1-{\text{Mask}}_{ijt}$’ aims to keep damaged points and remove other irrelevant points.

## 5. Experiment

### 5.1. Datasets and Settings

In this study, the experimental data are collected from two urban expressways named Yan’an and Neihuan of Shanghai, China in 2011. Figure 6 shows the map of these two elevated highways, which are important parts of Shanghai’s urban transportation network and effectively increase the traffic capacity.

On each elevated highway, there is a loop detector every 400 m. The detector collects and stores traffic data at its location every 5 min, including flow, speed, and occupancy. There are 35 and 72 detectors on the Yan’an and Neihuan elevated highways, respectively. These detectors collect 288 time points in a day.

To use the correlation between time and space, we first convert the raw data collected from loop detectors into daily spatio–temporal matrices. However, there may exist errors in the spatio–temporal matrix, because of the inevitable damage of the detector and storage. Therefore, these data needs further process. Firstly, we use neighbour average filtering to handle the invalid value ‘0’ in the matrix. Secondly, we choose to use data collected from 7 a.m. to 10 p.m. for experiments, because some loop detectors may be maintained at night and fail to collect traffic data. Lastly, in terms of the Yan’an elevated highway, we only have data from 361 days to make the data set, due to the lack of data from 20 March to 23 March. After making these processes, to simulate the traffic data damage, we use the two masks mentioned in Section 4.2. A value of 0 in the mask indicates that data is damaged. Multipling the mask and the original speed space–time matrix point-by-point yields the damaged speed matrix. Figure 7 shows the strip damage. It may appear when a detector fails and lasts for a period of time. Figure 8 illustrates the discrete damage. This situation may occur when the transient detector fails or data is lost during storage. The damaged speed space–time matrix, the flow matrix, and the occupancy matrix together are taken as the input, and the label is the complete speed space–time matrix. All the matrices for the whole of 2011 constitute the basic data set of each elevated highway. To evaluate the performance of our proposed model, we randomly select 36 samples as the test set for each data set, and the remaining samples are regarded as the training set. For Yan’an and Neihuan elevated highways, their respective training sets include 325 and 329 samples.

The experiments are conducted on a server with i7-5820K CPU, 48 GB memory and NVIDIA GeForce GTX1080 GPU. The proposed model is implemented on the TensorFlow framework of deep learning, whose parameter configuration is shown in Table 2 and Table 3. Note that, the parameters and network structure of the two elevated highways are the same. The step size of all convolution kernels is set as 1 × 1. The learning rate of both the generator and the discriminator is 0.0001, and the total number of iterations of our network is 10,000.

The numbers of training samples of Yan’an and Neihuan expressways are 325 and 329, respectively. These samples are considered as inputs to train the proposed GAGAN model by minimizing the loss function. Once the training process is terminated, the learned weight matrices will be immediately saved. There are 36 randomly selected test samples for every elevated highway, each sample is delivered to the saved generator of GAGAN to yield a predicted speed matrix by forward calculation. Combing the predicted speed matrix and its associated mask, missing speed values can be recovered.

### 5.2. Results and Analysis

We first visualize speed matrices as heat maps, which reveal the traffic speed values in a whole day, to demonstrate the repaired results of our model. In each heat map, the x-axis represents the time series of one day, and the y-axis indicates the position of these detectors. In addition, the different values of speed are represented with different colors. The darker the color is, the smaller the speed value is. Figure 9 includes the heat maps of Yan’an elevated highway with strip damage. From left to right are the mask, the damaged speed matrix, the ground truth and its corresponding repaired speed matrix on the 1st day of test set. Similarly, Figure 10 shows the results of 8th day of the test set with strip damage. Figure 11 contains the results of 31st day of the test set with discrete damage. Obviously, the repaired speed data of our model are very close to the ground truth for the Yan’an elevated highway with both strip damage and discrete damage. Then we conduct the same experiments on the Neihuan elevated highway, and the results are depicted in Figure 12, Figure 13 and Figure 14. It is also illustrated that our proposed method achieves a close result to the ground truth for the Neihuan elevated highway with both strip damage and discrete damage.

In this paper, we use L1 loss and L2 loss to evaluate the repair performance. The L1 loss indicates the average absolute error (MAE) of the damaged location, and the L2 loss is used to measure the mean squared error (MSE). The formulas of L1 loss and L2 loss are defined as
(23)$$L_{1}=\frac{1}{v}\sum_{j=1}^{v}\left(\frac{1}{u}\sum_{i=1}^{u}\left|y_{ij}-{\hat{y}}_{ij}\right|\right)$$
(24)$$L_{2}=\frac{1}{v}\sum_{j=1}^{v}\left(\frac{1}{u}\sum_{i=1}^{u}{\left(y_{ij}-{\hat{y}}_{ij}\right)}^{2}\right)$$
where $y_{ij}$ means the true value of the $i_{th}$ damaged point in the speed matrix of $j_{th}$ test sample, ${\hat{y}}_{ij}$ is the corresponding repaired value, u indicates the number of damaged points in a recovered speed matrix and v denotes the number of samples in the test set.

To improve the efficiency of the proposed model, we compare our method with CNNBranch3 [27], CNN3 [28], CNN1 [22] and CNNBranch3_fc, they all have the GAN architecture. In order to prove the advantages of geometric algebra convolution, CNNBranch3 is used as its comparative experiment. The difference between CNNBranch3 and our model is only the convolutional layer of the generator. CNNBranch3 uses traditional scalar convolution, while our model has geometric algebra convolution. Compared with CNNBranch3, which takes a multibranch structure to process inputs with parameters of F, S and O, CNN3 simply uses 3D convolution to process inputs with these three parameters. In order to prove the influence of parameter correlation on the repair effect, CNN1 was employed for comparison. It only takes the damaged speed parameter as the input; flow and occupancy are not included. CNN1 also uses scalar convolution. For the last CNNBranch3_fc model, compared with the CNNBranch3 model, the deconvolution layer of the generator is replaced with a fully connected layer, and the other modules remain unchanged to prove the importance of decoding high-dimensional features.

In the experiment, the generator of CNNBranch3 contains three branches, which are used to process traffic, speed, and occupancy data. These three branches encode and decode the corresponding parameters. The results generated by the branches are merged, the repaired data can be obtained from the last deconvolution layer. The encoding part has three convolutional layers, and the decoding also includes three deconvolutional layers. The generator of CNN3 employs 3D convolution to process the three traffic parameters, without merging feature maps such as branches, and other module structures are the same as that of CNNBranch3. The input of the CNN1 generator is the impaired speed matrix, which is also composed of three convolutional layers and three deconvolutional layers. The structure of CNNBranch3_fc and CNNBranch3 is basically the same, except that the three deconvolution layers in the generator is replaced with three fully connected layers.

Curves of Figure 15 and Figure 16 demonstrate the recovered values and their corresponding ground truth of the Yan’an and Neihuan elevated highways for some detectors. More specifically, we randomly selected six diagrams of each highway, which represent the speed values of different detectors on different days. In these subgraphs, the blue solid line denotes the ground truth, the yellow solid line represents the repaired value of our model, and other curves with different colors indicate the results of baseline methods. It can be seen from these figures that the repaired results generated by our model are the closest to the ground truth.

Finally, we use strip mask and show the results compared with baseline methods in Table 4 and Table 5. It can be found that the proposed model achieves the lowest error among all of the methods. More specifically, CNN1 and CNNBranch3_fc perform the worst, as CNN1 does not consider the correlation of traffic parameters, and CNNBranch3_fc cannot decode the extracted features effectively. The performance of CNN3 is better than the previous two, because it makes full use of the excellent decoding ability of parameter correlation and deconvolution. Compared with CNN3, CNNBranch3 performs feature extraction on each parameter separately, so that all parameters can be fused after they have been fully learned, and the performance is better. Our model shows the best performance due to the use of geometric algebra convolution. For the Yan’an elevated highway, L1 and L2 indicators produced by our method are 3.264% and 0.259%, which are the lowest of all listed methods. Compared with the presented models, L1 and L2 measures of our approach for the Neihuan elevated highway have the values of 2.616% and 0.180%, which are still the smallest.

In order to further verify the generalization ability, we conduct a comparative experiment with different degrees of damage in the case of discrete damage. The ratios of damaged area to total area are 10%, 20%, 30%, 40% and 50%. Figure 17 demonstrates that as the degree of damage increases, the performance of the models also declines within a reasonable range. However, the proposed method still performs the best, which proves the robustness of our model.

In this section, various experiments are conducted to evaluate the robustness of our model and to compare with other state-of-the-art work. It is illustrated that our method outperforms CNNBranch3 [24], CNN3 [25], CNN1 [19] and CNNBranch3_fc. In addition, the proposed approach performs well with both strip damage and discrete damage on two highways. Specifically, in the case of discrete damage, the generalization ability of our model with different degrees of damage is also proved. The performance of the proposed GAGAN model greatly contributes to the joint learning of correlation between high-dimensional traffic parameters.

## 6. Conclusions

In this paper, we propose a geometric-algebra-based generative adversarial network to deal with the important task of repairing missing traffic speed data. The original traffic data which include speed, flow and occupancy are first processed as spatial–temporal matrices. To make full use of the correlation between different traffic parameters, the speed, flow and occupancy data are embedded in the geometric algebraic framework to form multivectors and used as the input of the proposed model. The geometric algebra convolution module in the generator encodes high-dimensional data and enables efficient joint learning of multidimensional traffic parameters. The deconvolution module in the generator decodes the extracted features and generates recovered traffic speed matrix. In the proposed model, the generator loss function takes into account the feedback information from the discriminator, the global and local traffic speed data characteristics at the same time. The discriminator based on the multichannel convolutional network makes the repair value more realistic. Traffic data obtained from the elevated highway loop detectors are used to evaluate the performance of the proposed method. Experimental results show that our approach outperforms the state-of-the-art work and can effectively recover missing traffic speed data in a robust way.

## Figures and Tables

**Figure 1 sensors-22-02744-f001:**
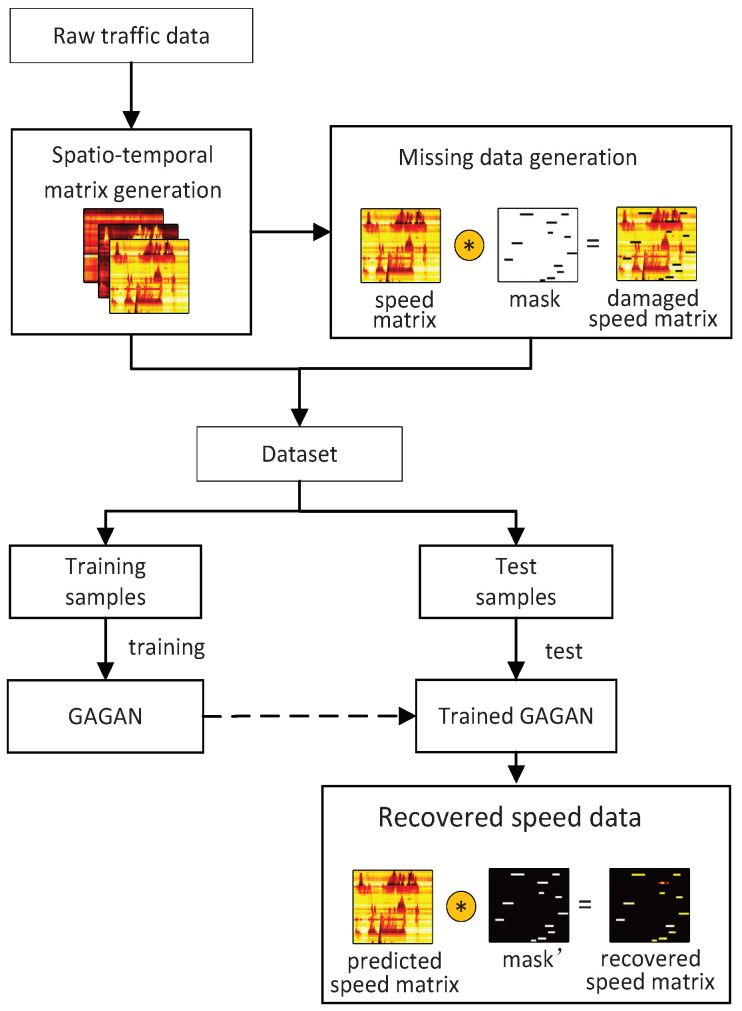
The system architecture of traffic speed imputation using GAGAN.

**Figure 2 sensors-22-02744-f002:**
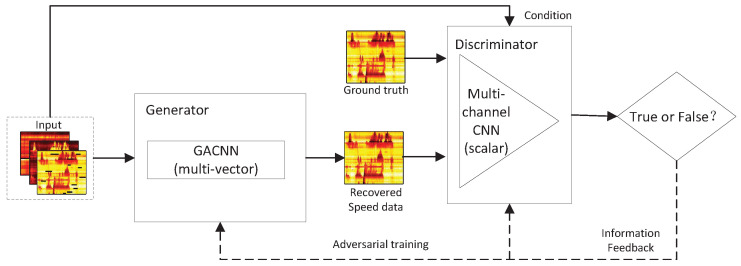
The geometric algebra based generative adversarial network (GAGAN).

**Figure 3 sensors-22-02744-f003:**
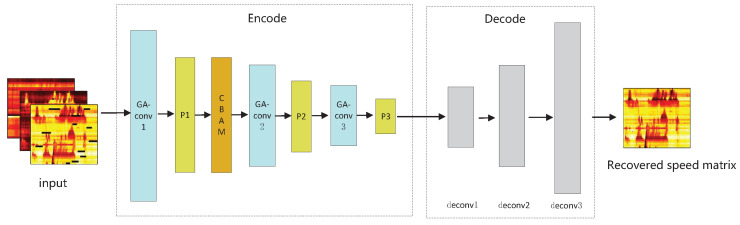
The generator of GAGAN.

**Figure 4 sensors-22-02744-f004:**
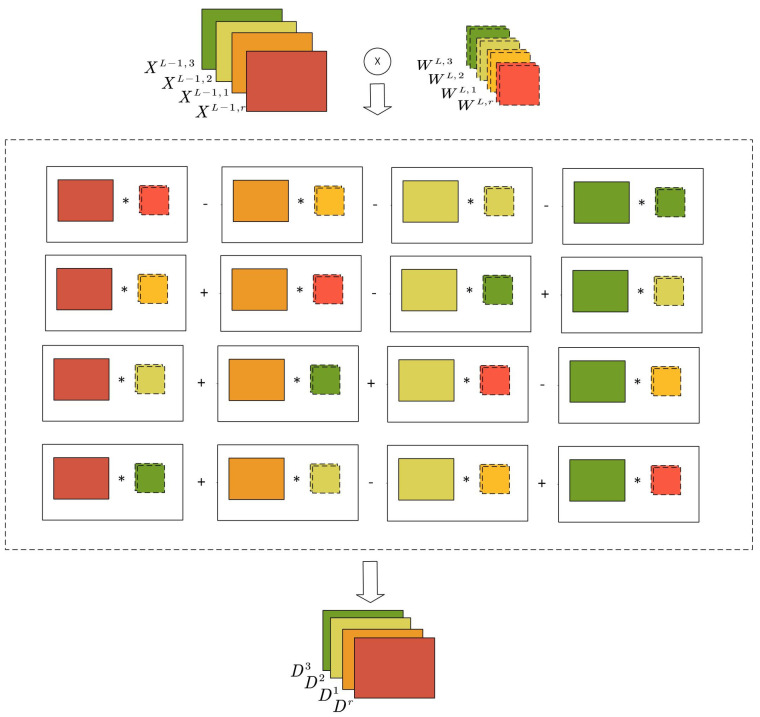
The implementation of geometric algebra convolutional layer.

**Figure 5 sensors-22-02744-f005:**
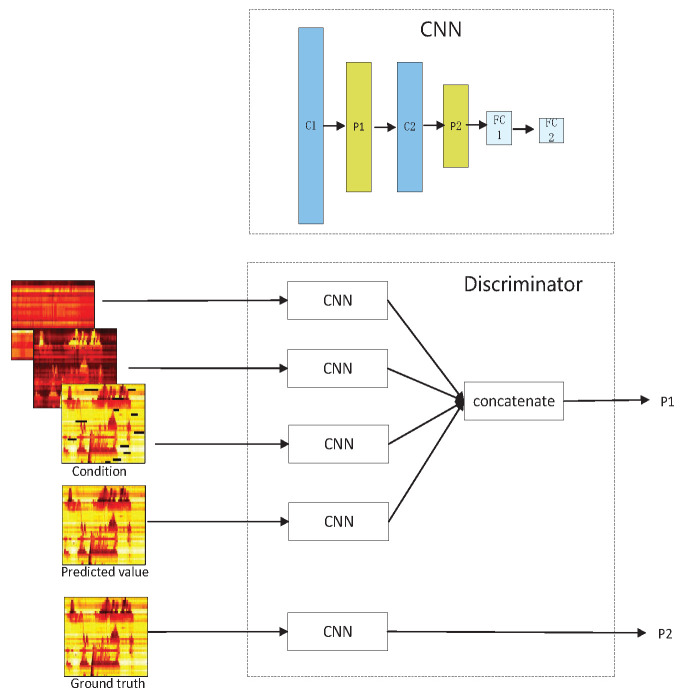
The structure of discriminator of GAGAN.

**Figure 6 sensors-22-02744-f006:**
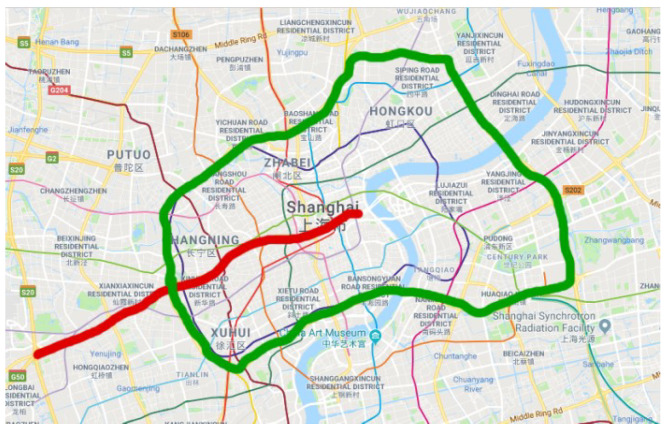
Marking of two elevated highways. Red and green bold lines mark Yan’an elevated highway and Neihuan elevated highway, respectively.

**Figure 7 sensors-22-02744-f007:**
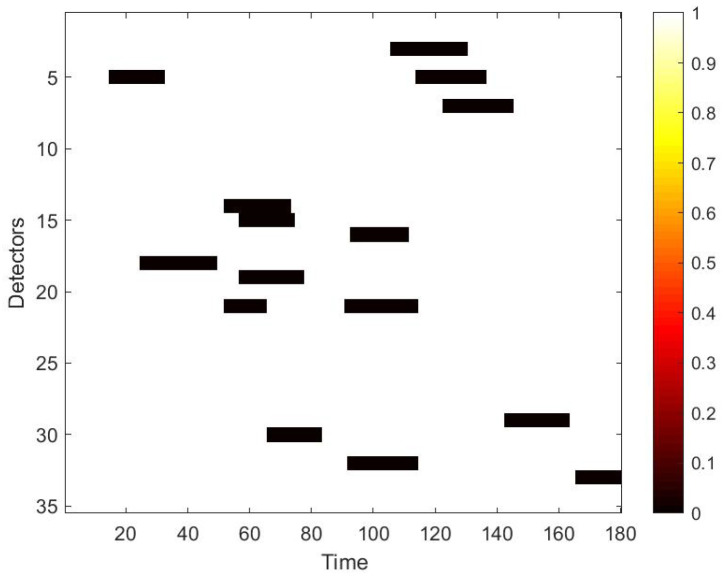
Mask used to simulate strip damage.

**Figure 8 sensors-22-02744-f008:**
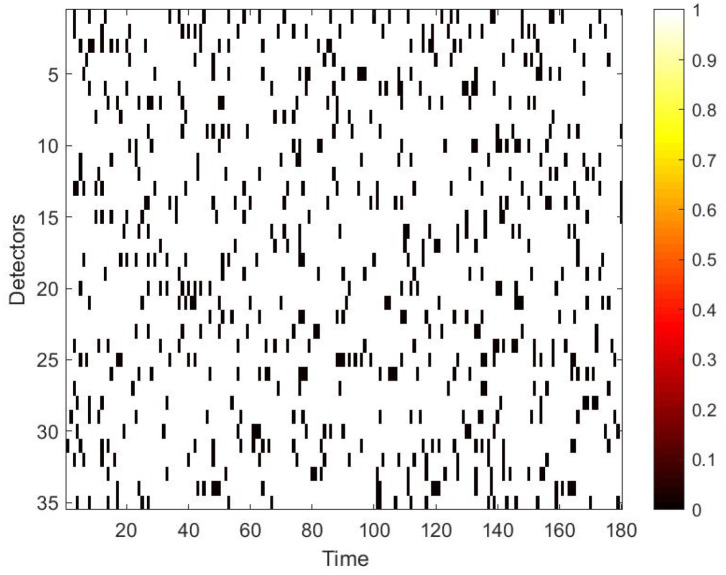
Mask used to simulate discrete damage.

**Figure 9 sensors-22-02744-f009:**
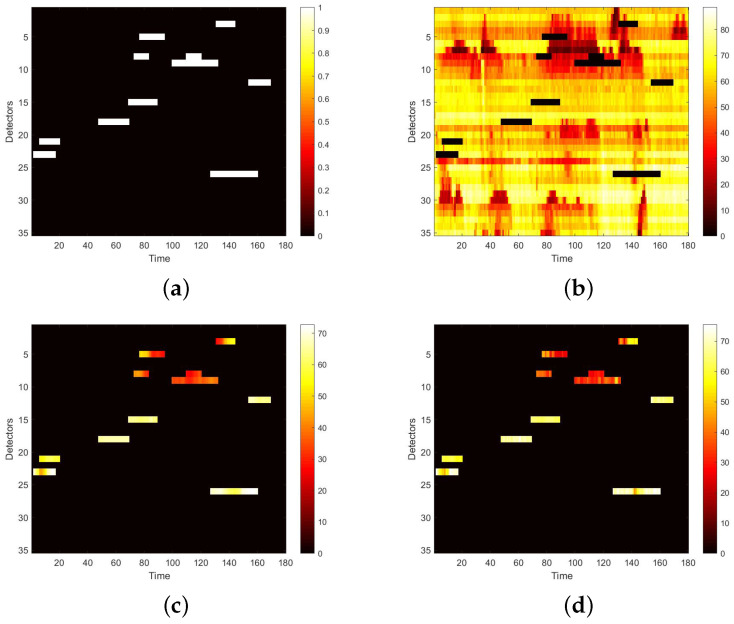
Speed matrices of Yan’an elevated highway (the 1st day of testset) with strip damage visualized as heat maps. (**a**) Mask matrix visualized as a heat map. (**b**) Damaged speed matrix visualized as a heat map. (**c**) Repaired speed matrix visualized as a heat map. (**d**) Real speed matrix visualized as a heat map.

**Figure 10 sensors-22-02744-f010:**
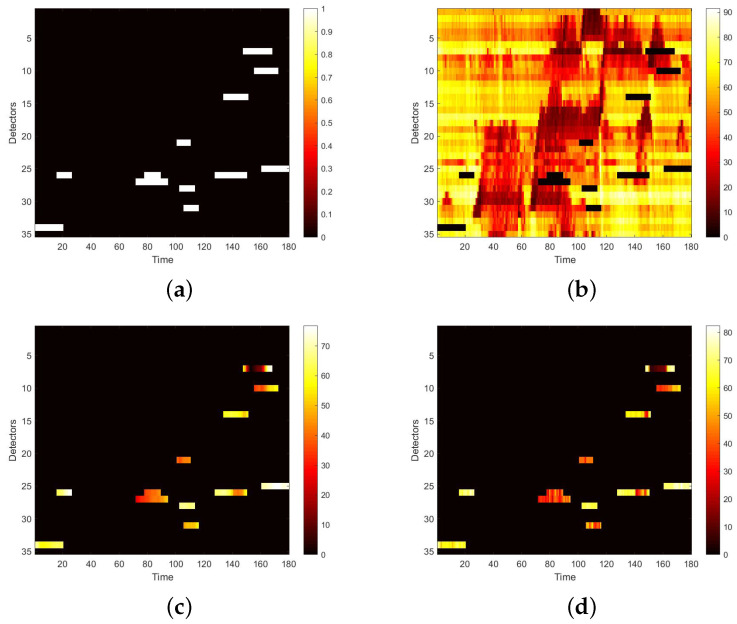
Speed matrices of Yan’an elevated highway (the 8th day of testset) with strip damage visualized as heat maps. (**a**) Mask matrix visualized as a heat map. (**b**) Damaged speed matrix visualized as a heat map. (**c**) Repaired speed matrix visualized as a heat map. (**d**) Real speed matrix visualized as a heat map.

**Figure 11 sensors-22-02744-f011:**
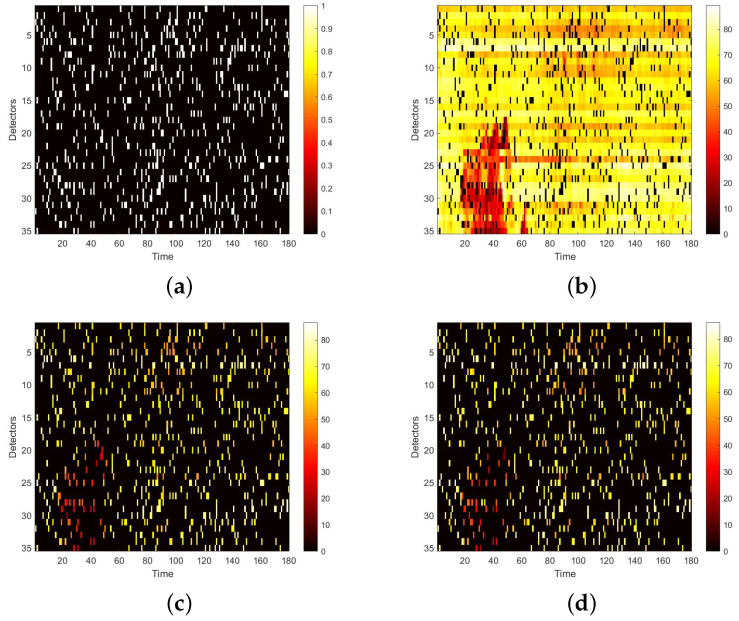
Speed matrices of Yan’an elevated highway (the 31st day of testset) with discrete damage visualized as heat maps. The ratio of damaged area to total area is 10%. (**a**) Mask matrix visualized as a heat map. (**b**) Damaged speed matrix visualized as a heat map. (**c**) Repaired speed matrix visualized as a heat map. (**d**) Real speed matrix visualized as a heat map.

**Figure 12 sensors-22-02744-f012:**
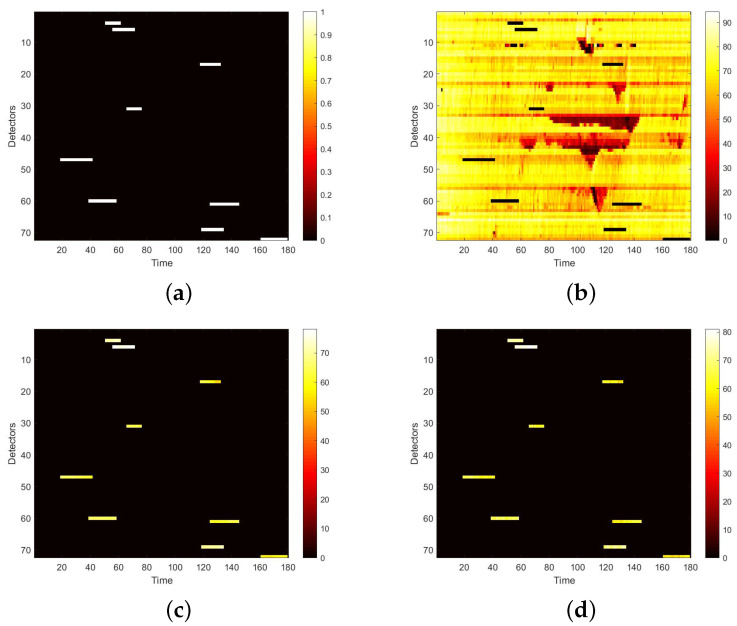
Speed matrices of Neihuan elevated highway (the 6thday of testset) with strip damage visualized as heat maps. (**a**) Mask matrix visualized as a heat map. (**b**) Damaged speed matrix visualized as a heat map. (**c**) Repaired speed matrix visualized as a heat map. (**d**) Real speed matrix visualized as a heat map.

**Figure 13 sensors-22-02744-f013:**
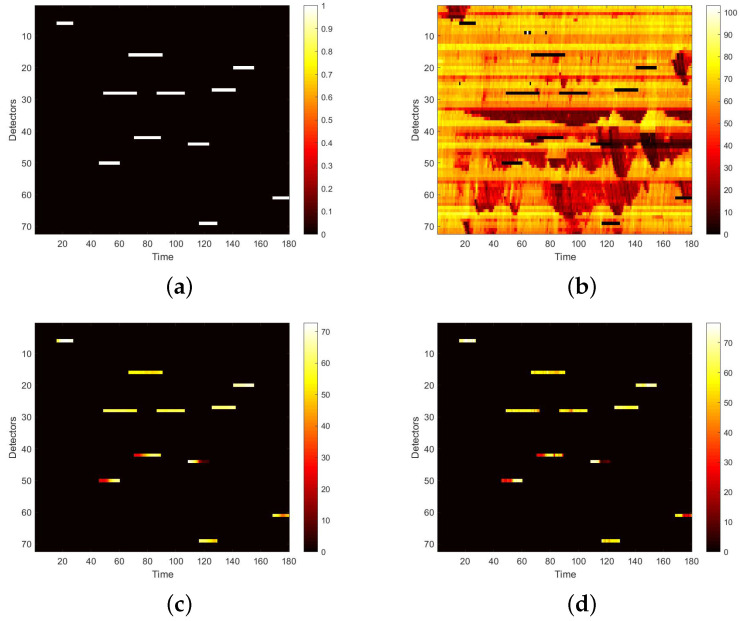
Speed matrices of Neihuan elevated highway (the 23rd day of testset) with strip damage visualized as heat maps. (**a**) Mask matrix visualized as a heat map. (**b**) Damaged speed matrix visualized as a heat map. (**c**) Repaired speed matrix visualized as a heat map. (**d**) Real speed matrix visualized as a heat map.

**Figure 14 sensors-22-02744-f014:**
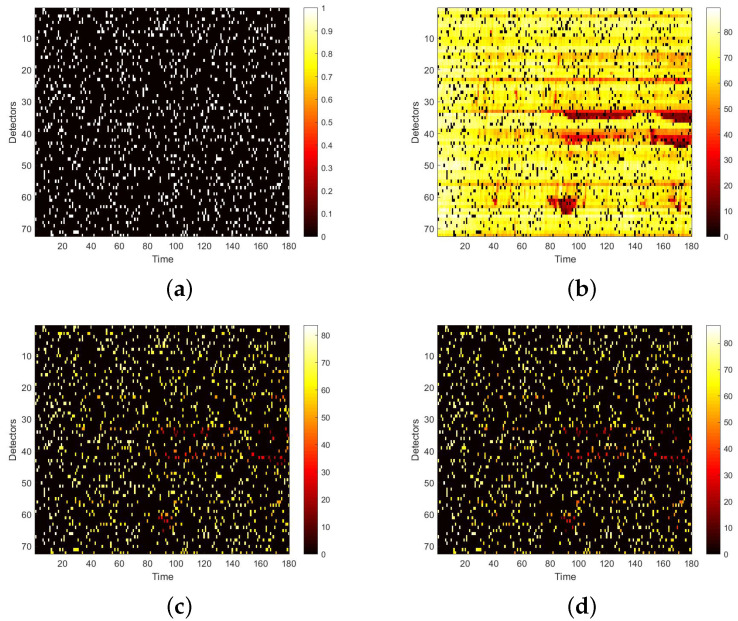
Speed matrices of Neihuan elevated highway (the 36th day of testset) with discrete damage visualized as heat maps. The ratio of damaged area to total area is 10%. (**a**) Mask matrix visualized as a heat map. (**b**) Damaged speed matrix visualized as a heat map. (**c**) Repaired speed matrix visualized as a heat map. (**d**) Real speed matrix visualized as a heat map.

**Figure 15 sensors-22-02744-f015:**
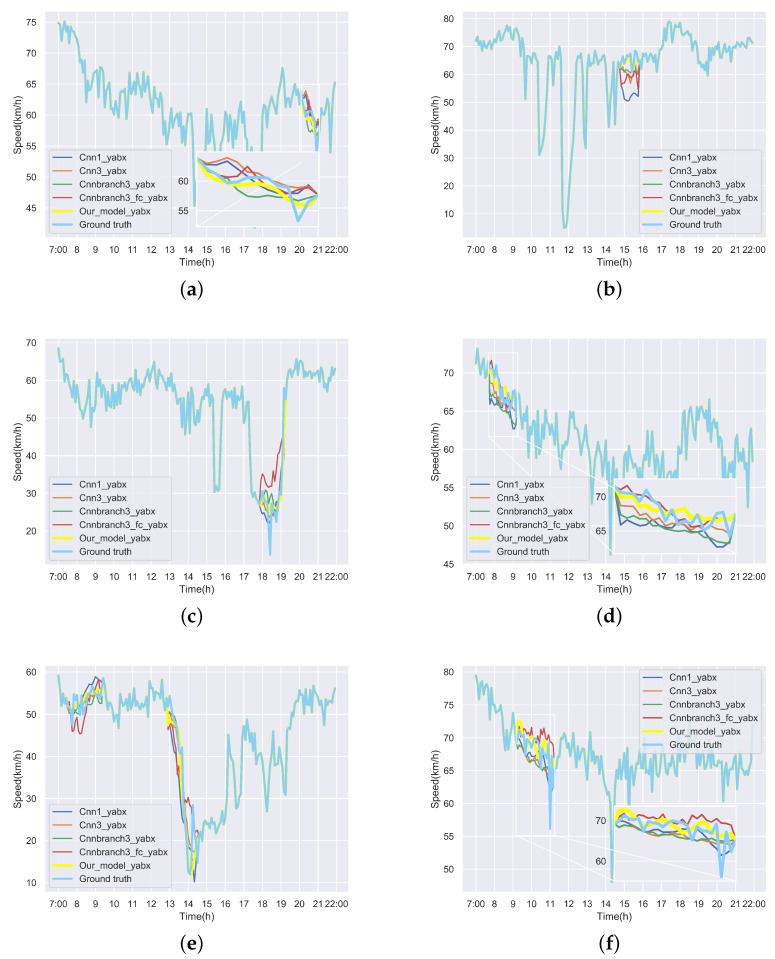
Repaired speed curves and corresponding ground truth of six loop detectors of Yan’an elevated highway. (**a**) The repaired values and the ground truth of the 10th loop detector on the Yan’an elevated highway in the 2nd test day. (**b**) The repaired values and the ground truth of the 35th loop detector on the Yan’an elevated highway in the 3rd test day. (**c**) The repaired values and the ground truth of the 8th loop detector on the Yan’an elevated highway in the 7th test day. (**d**) The repaired values and the ground truth of the 11th loop detector on the Yan’an elevated highway in the 10th test day. (**e**) The repaired values and the ground truth of the 19th loop detector on the Yan’an elevated highway in the 32nd test day. (**f**) The repaired values and the ground truth of the 15th loop detector on the Yan’an elevated highway in the 35th test day.

**Figure 16 sensors-22-02744-f016:**
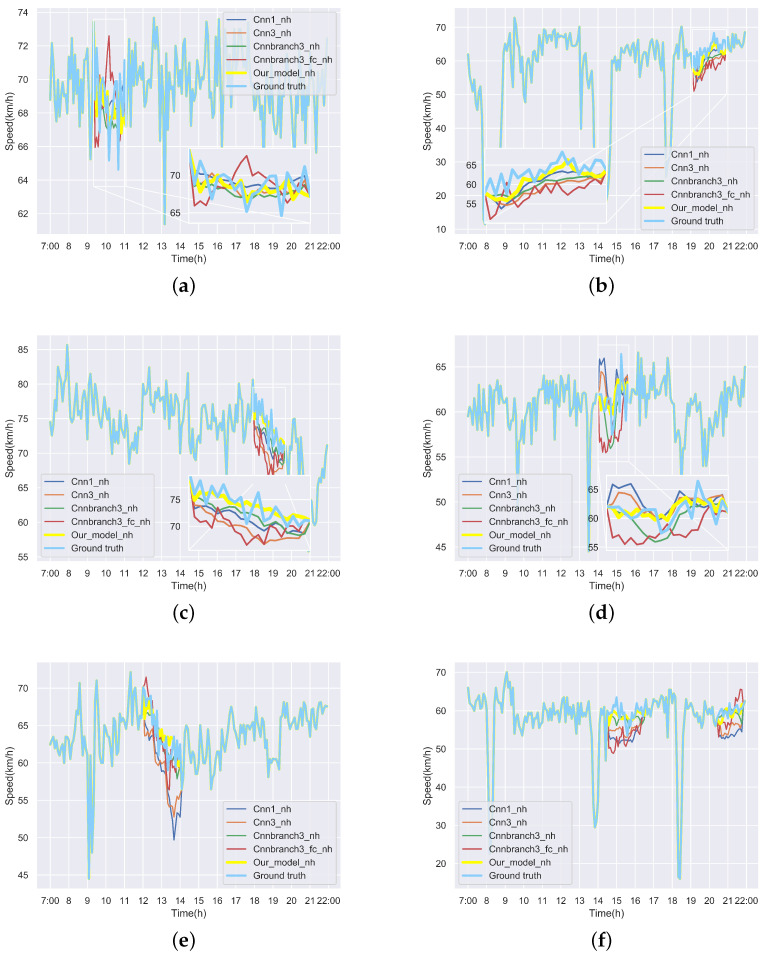
Repaired speed curves and corresponding ground truth of six loop detectors of Neihuan elevated highway. (**a**) The repaired values and the ground truth of the 7th loop detector on the Neihuan elevated highway in the 2nd test day. (**b**) The repaired values and the ground truth of the 24th loop detector on the Neihuan elevated highway in the 10th test day. (**c**) The repaired values and the ground truth of the 14th loop detector on the Neihuan elevated highway in the 12th test day. (**d**) The repaired values and the ground truth of the 10th loop detector on the Neihuan elevated highway in the 15th test day. (**e**) The repaired values and the ground truth of the 32nd loop detector on the Neihuan elevated highway in the 21st test day. (**f**) The repaired values and the ground truth of the 31st loop detector on the Neihuan elevated highway in the 27th test day.

**Figure 17 sensors-22-02744-f017:**
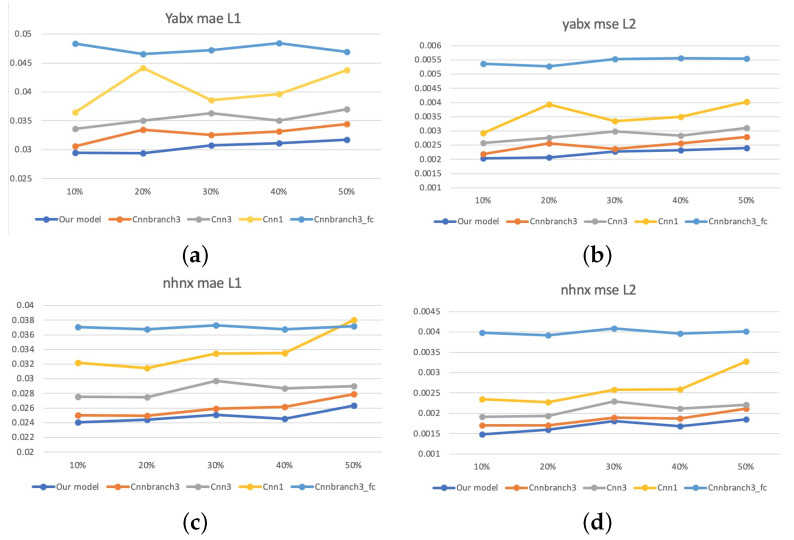
Results of different levels of damage on two elevated highways. (**a**) L1 of different levels of damage on Yan’an elevated highways. (**b**) L2 of different levels of damage on Yan’an elevated highways. (**c**) L1 of different levels of damage on Neihuan elevated highways. (**d**) L2 of different levels of damage on Neihuan elevated highways.

**Table 1 sensors-22-02744-t001:** The geometric product of basis elements of $R_{3}$.

	1	$e_{1}$	$e_{2}$	$e_{3}$	$e_{12}$	$e_{23}$	$e_{31}$	$e_{123}$
1	1	$e_{1}$	$e_{2}$	$e_{3}$	$e_{12}$	$e_{23}$	$e_{31}$	$e_{123}$
$e_{1}$	$e_{1}$	1	$e_{12}$	$e_{13}$	$e_{2}$	$e_{123}$	$-e_{3}$	$e_{23}$
$e_{2}$	$e_{2}$	$-e_{12}$	1	$e_{23}$	$-e_{1}$	$e_{3}$	$-e_{123}$	$-e_{13}$
$e_{3}$	$e_{3}$	$-e_{13}$	$-e_{23}$	1	$-e_{1}$	$e_{123}$	$-e_{1}$	$e_{12}$
$e_{12}$	$e_{12}$	$-e_{2}$	$e_{1}$	$e_{123}$	−1	$e_{13}$	$e_{23}$	$-e_{3}$
$e_{23}$	$e_{23}$	$e_{123}$	$-e_{3}$	$e_{2}$	$-e_{13}$	−1	$-e_{12}$	$-e_{1}$
$e_{31}$	$e_{31}$	$e_{3}$	$e_{123}$	$e_{1}$	$-e_{23}$	$e_{12}$	−1	$-e_{2}$
$e_{123}$	$e_{123}$	$e_{23}$	$-e_{13}$	$e_{12}$	$-e_{3}$	$-e_{1}$	$-e_{2}$	−1

**Table 2 sensors-22-02744-t002:** Parameter configuration of the generator of GAGAN.

Layers	Name	Description
1	GA-conv Layer1	32 kernels of size 3 × 3 × 1
2	Pooling1	kernels of size 2 × 2
3	CBAM	attention module
4	GA-conv Layer2	64 kernels of size 3 × 3 × 32
5	Pooling2	kernels of size 2 × 2
6	GA-conv Layer3	64 kernels of size 3 × 3 × 64
7	Pooling3	kernels of size 2 × 2
8	deconv Layer1	64 kernels of size 3 × 3 × 64
9	deconv Layer2	64 kernels of size 3 × 3 × 32
10	deconv Layer3	32 kernels of size 3 × 3 × 1

**Table 3 sensors-22-02744-t003:** Parameter configuration of the discriminator of GAGAN.

Layers	Name	Description
1	convolution1	32 kernels of size 5 × 5 × 31
2	pooling1	kernels of size 2 × 2
3	convolution2	64 kernels of size 3 × 3 × 32
4	pooling2	kernels of size 2 × 2
5	FC1	128 neuron nodes
6	FC2	1 neuron nodes

**Table 4 sensors-22-02744-t004:** Comparison results for Yan’an elevated highway.

Models	L1	L2
Our model	0.03264	0.00259
Cnnbranch3	0.03495	0.00271
Cnn3	0.03960	0.00357
Cnn1	0.05014	0.00573
Cnnbranch3fc	0.05198	0.00636

**Table 5 sensors-22-02744-t005:** Comparison results for Neihuan elevated highway.

Models	L1	L2
Our model	0.02616	0.00180
Cnnbranch3	0.02995	0.00254
Cnn3	0.03741	0.00312
Cnn1	0.04830	0.00490
Cnnbranch3fc	0.04034	0.00399

## Data Availability

Not applicable.

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
