# Peer review of "Traffic-Data Recovery Using Geometric-Algebra-Based Generative Adversarial Network"

_sensors, 2022, doi:10.3390/s22072744_

Round 1

Reviewer 1 Report

Read " Reviewer suggestions.docx" for details.

Reviewer 2 Report

This paper deals with an exciting topic. The article has been read carefully, and some minor issues have been highlighted in order to be considered by the author(s).

#1 What is the motivation of this paper?

#2 What is the contribution and novelty of this paper?

#3 What is the advantage of this survey paper?

#4 Which evaluation metrics did you used for comparison?

#5 It would be good if security or similar domains for the deep neural network would be reflected in the related work such as Optimized Adversarial Example with Classification Score Pattern Vulnerability Removed.

Reviewer 3 Report

This paper describes an aproach for missing road traffic data reconstruction using geometric-algebra-based generativ adversial network.

The entire approach is well described and seems to be sound.

The structure of the paper is relatively good. Nevertheless, the state-of-the-art, which is now part of Section 1 should be moved to a new Section 2. It is better to have a separate state-of-the-art section than to have it merged with the introduction, because the introduction should be relatively short to give the reader the information, what is described in the paper and why. The necessity of third-level headings in paper of this length is also questionable, though not necessarily wrong.

The references in the paper seem to be relevant and up to date.

The figures are appropriate and of sufficient quality.

The English is good, the amount of typos and errors is low. Proofreading by a grammar-skilled native speaker can still be useful.

Reviewer 4 Report

The authors describe in the manuscript the traffic data recovery by geometric algebra based generative adversarial network method.
The issue under consideration plays an important role in traffic prediction, congestion judgment, road network planning and other fields. Authors propose the geometric algebra based generative adversarial network to repair the missing traffic data by learning the correlation of multidimensional traffic parameters. Experimental results demonstrate that our method can effectively repair missing traffic data in a robust way and has better performance when compared with the state-of-the-art work. These are interesting and serious scientific achievements. There are also considerable inaccuracies in the article.
1. There are no comments in the manuscript regarding the training of this class of models with the use of the entropy function. Could such models really not be able to estimate the missing data?
2. In the literature on the subject, some sources have titles written in capital letters, and others in small letters.
3. Quotations are missing spaces: date [xx] -> date [xx]
To summarize, the above comments can be corrected by the authors themselves, and after taking them into account, the manuscript will be a coherent description of this interesting algorithm.

Round 2

Reviewer 2 Report

I recommend the acceptance.